# Defense against Backdoor Attacks via Identifying and Purifying Bad Neurons

## Abstract

Recent studies reveal the vulnerability of neural networks to *backdoor attacks*. By embedding backdoors into the hidden neurons with poisoned training data, the backdoor attacker can override normal predictions of the victim model to the attacker-chosen ones whenever the backdoor pattern is present in a testing input. In this paper, to mitigate public concerns about the attack, we propose a novel backdoor defense via identifying and purifying the backdoored neurons of the victim neural network. Specifically, we first define a new metric, called *benign salience*. By combining the first-order gradient to retain the connections between neurons, benign salience can identify the backdoored neurons with high accuracy. Then, a new Adaptive Regularization (AR) mechanism is proposed to assist in purifying these identified bad neurons via fine-tuning. Due to the ability to adapt to different magnitudes of parameters, AR can provide faster and more stable convergence than the common regularization mechanisms in neuron purifying. Finally, we test the defense effect of our method on ten different backdoor attacks with three benchmark datasets. Experimental results show that our method can decrease the attack success rate by more than 95% on average, which is the best among six state-of-the-art defense methods.

## 1 Introduction

The brilliant feat of neural network (NN) make it a focal point of many attacks, one of the most threatening among which is the *backdoor attack* (Liu et al., 2017; Barni et al., 2019). By mixing poisoned data into the training set, backdoor attack can control the victim NN to output attacker-chosen predictions for triggered inputs, while hardly disturb the predictions of normal inputs. Moreover, the triggers crafted by the attacker can be only several blocks of pixels (Li et al., 2020b) or even the invisible noises (Zhong et al., 2020), which makes the attack notoriously perilous in applications.

Based on the findings in (Gu et al., 2017), backdoor attacks can succeed because the neurons that memorize trigger patterns, often called *bad neurons*, only keep a strong connection with the triggers but rarely react to normal features. Naturally, to defend the attack, a direct idea is to remove or mitigate the effect of bad neurons, e.g., model-pruning or fine-tuning (Gu et al., 2017). However, the existing defense methods are mainly flawed from two perspectives.

**Neuron Evaluation.** To evaluate which neurons are backdoored, current approaches are mostly based on the activation magnitude (AM) of neurons (Gu et al., 2017). Neurons with low AMs about normal inputs are deemed to be "bad", and pruned during the defense stage. Despite being intuitive and easy to apply, such a metric sometimes leads to the over-pruning of good neurons as it may ignore the connections between neurons. For instance, consider a case where a clean neuron $\mathcal{N}$ in the network has low AMs but has very high weights on its downstream neurons. $\mathcal{N}$ has a strong positive effect on the final predictions but is still judged to be bad based on the above rule.

**Neuron Purifying.** As bad neurons are marked in the evaluation stage, the next step is to purify or prune them. Currently, most pruning based defense methods choose to roughly remove bad neuron from the backdoored network. However, as pointed out by prior works (Liu et al., 2018), some bad neurons can also be related to the predictions of normal data. Roughly removing these neurons can easily cause the performance degradation of the target model.

In this paper, we propose a novel backdoor defense method (called WIPER) to overcome the two flaws. Specifically, instead of utilizing AM, we define a more effective metric called *benign salience* (BS, Section 4.1) to evaluate the importance of neurons. Compared to AM, BS retains the connections between neurons through the first-order gradient, hence bad neurons can be identified more accurately. Then, after identifying bad neurons, our method chooses to fine-tune but not directly prune them in the neuron purifying stage (Section 4.2) to avoid unexpected performance degradation. Through a newly designed adaptive piece-wise regularization mechanism, our fine-tuning method can be far more effective in mitigating the network's attention on trigger patterns than the existing fine-tuning method (Truong et al., 2020). Our contributions are summarized in four folds:

- We propose WIPER, a novel backdoor defense method that combine the advantages of both model pruning and fine-tuning to identify and purify backdoored neurons.

- We design a new metric BS to mark the bad neurons whose attention is misled to the backdoor trigger patterns. Since the connections between neurons are reserved via the first-order gradient, defenders can use BS to distinguish bad neurons with higher accuracy than the commonly used metrics, e.g., AM.

- We develop a new type of regularization adaptive regularization (AR). Compared to the common regularization, AR can better accelerate and stabilize the purifying process of bad neurons by adaptively adjusting the penalty degree to different magnitudes of parameters.

- We conduct extensive experiments on benchmark datasets to validate the effectiveness of WIPER. The result shows that our method outperforms state-of-the-art defenses significantly on both attack success rate decreasing and model performance maintenance.

## 2 RELATED WORKS

**Backdoor attack.** A typical backdoor attack is done by injecting a small volume of poison data crafted with the attacker-chosen triggers into the training set. To maintain the stealthiness of backdoor attacks, the triggers used by attackers are various. For instance, the trigger used in (Li et al., 2020b) was a simple rect with the resolution of $3 \times 3$. Blend attack (Chen et al., 2017) adopted common life devices, e.g. glasses, as a trigger so as to evade the inspection of humans. Moreover, the authors in (Zhong et al., 2020) attempted to design a human-imperceptible yet effective noise. The clean-label attack (Shafahi et al., 2018) made the natural features in images difficultly learnable so that the model was forced to only rely on the trigger to correctly classify without modifying the label. More recently, similar to clean-label attack, sinusoidal signal attack (Barni et al., 2019) designed an easily-learned trigger to conduct the backdoor attack. WIPER is designed to provide an effective to defend the above-mentioned attacks and ensure model security in applications.

**Backdoor defense.** Backdoor defenses can be roughly divided into two categories: detection-based methods (Wang et al., 2019; 2020; Xu et al., 2021) that aim to detect whether a neural network is backdoored, and purifying-based methods (Zhao et al., 2019; Truong et al., 2020; Yoshida & Fujino, 2020; Li et al., 2021a) that try to remove the backdoor while maintaining the performance of the target model. Up to now, the detection-based method has been well developed and many remarkable works (Gao et al., 2019; Wang et al., 2020; Xu et al., 2021) achieve quite a high detection rate. Thus, this paper mainly focuses on the bad neuron purifying to remove the backdoor. Inspired by the fact that the bad neurons were dormant with the presence of clean data, fine-pruning (Liu et al., 2018) removed the backdoor by erasing the neurons with activation values below a certain threshold. However, the effectiveness of fine-pruning heavily depends on the quality of holding data, and it can be easily evaded by some state-of-the-art attacks, such as TrojanNN (Liu et al., 2017). In (Truong et al., 2020), the authors suggested that referring to catastrophic forgetting (Delange et al., 2021), fine-tuning the model with some clean data was a simple yet effective method to remove the backdoor. Similar to fine-pruning, fine-tuning also need high-quality clean data to mitigate the effect of bad neurons. Recently, knowledge distillation (Yoshida & Fujino, 2020; Li et al., 2021a) was proposed to achieve backdoor defense by distilling clean knowledge from the infected model to a fresh model but failed to defend TrojanNN (Liu et al., 2017) due to the lack of consideration for its special trigger attention mechanism. Recently, a novel work, AI-Lancet (Zhao et al., 2021), was proposed to locate bad neurons by doing comparative experiments with triggered inputs and clean inputs whose trigger-pasted regions were cropped. Such an idea achieves competitive defense

performance against pixel-level backdoor attacks, however, fails to respond to the state-of-the-art attacks that uses crafted noises as triggers to cover the whole input image.

# 3 PROBLEM DESCRIPTION

**Problem Setting.** We consider the typical backdoor attack scenario as discussed in prior works (Gu et al., 2017; Liu et al., 2017; Barni et al., 2019; Li et al., 2020b). In the scenario, a model trainer is with the knowledge of the training set $D_t$ collected from multiple data providers and a self-owned clean set $D_c$. The trainer seeks to leverage $D_t$ to train a clean neural network $\mathcal{F}$. While, among the data providers, there exists at least a malicious one $\mathcal{A}$ that attempts to backdoor the trained model $\mathcal{F}$ by stealthily adding poisoned data into $D_t$.

**Attack Goal.** In the above scenario, the attacker $\mathcal{A}$ aims to complete the following goals.

- By crafting the testing inputs with specific triggers, $\mathcal{A}$ can mislead the backdoored network $\mathcal{F}'$ to output desired labels that are different from the ground-truth predictions (or predictions of a clean network $\mathcal{F}$).
- To maintain the stealthiness of the attack, $\mathcal{A}$ should make the backdoored network $\mathcal{F}'$ to perform similar to $\mathcal{F}$ as being given the clean inputs.

**Defense Goal.** As opposed to the attack goals, the defender, i.e., the model trainer who has full access to the internal architecture of the target model, aims to achieve the following two goals.

- The first is to erase the backdoors from $\mathcal{F}'$ and make the purified model perform correctly as being given with triggered inputs.
- To maintain utility, the purified model should perform similar to the original model trained with clean data (i.e., $D_t$ without poison data).

Note, to ensure the feasibility of the defense method in varying applications, e.g., federated learning (Li et al., 2020a), the defender is limited to only utilizing its self-owned validation set to achieve the defense goals.

# 4 APPROACH

In this section, we present the design details of WIPER. As illustrated in Algorithm 1 and Figure 1, WIPER mainly proceeds in two steps. The first is to find the bad neurons required to be cleansed by evaluating the importance of neurons. Then, a neuron-purifying strategy is leveraged to disable the effect of these bad neurons from the backdoored model.

## 4.1 NEURON IMPORTANCE EVALUATION

WIPER uses a novel metric called BS to evaluate the importance of neurons and distinguish bad neurons. As mentioned before, BS outperforms the conventional metric AM used in the existing pruning-based defense method (Liu et al., 2018) because instead of evaluating each neuron independently, BS additionally considers the connections between neurons in importance evaluation.

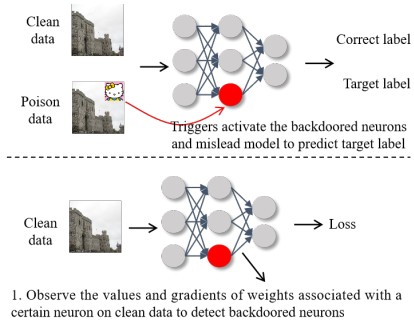

1. Observe the values and gradients of weights associated with a certain neuron on clean data to detect backdoored neurons
2. purify the corresponding weights of detected backdoored neurons

Figure 1: Backdoor attacks (top image) and WIPER (bottom image).

### 4.1.1 THE DEFINITION OF BS

In more details, the definition of BS basically follows two principles: 1) the importance of a neuron positively correlates to its contribution to the loss decreasing of the target network; 2) connections between neurons are established during the gradient backward propagation process in the training

---

**Protocol 1** WIPER

---

**Input:** $\mathcal{F}'$: the backdoored model; $D_c$: a set of clean data owned by the defender; $\mathbb{A}$: the set of neurons needs to be purified; $\alpha$: the hyperparameter to adjust the purifying magnitude; $\beta_0$: reserved ratio of neurons; $l$: the tag of the layer required to be purified; $n$: the total number of epoch of purifying; $\eta$: momentum accumulation factor.

**Output:** $\mathcal{F}$: the purified model.

 1: **# Neuron Importance Evaluation.**
 2: Compute the gradient of all neurons on the $l$-th layer over $D_c$.
 3: Evaluate the importance for the neurons on the $l$-th layer of $\mathcal{F}'$ based on Eq. 2 and Eq. 3.
 4: Select $\beta_0$ neurons with least importance (i.e., the lowest MBS) in the $l$-th layer and add them into $\mathbb{A}$.
 5: **# Neuron Purifying.**
 6: **for** each iteration $i = 0$ to $n$ **do**
 7:     Update $\beta_i \leftarrow \beta_0 \cdot (1 - \frac{i}{n})$.
 8:     **for** $x, y \in D_c$ **do**
 9:         Optimize $\mathcal{F}'$ based on $\mathcal{L}_{purifying}$.
10:         Update $MBS_i$ for each neuron in $l$-th layer and reset $\mathbb{A}$ that select $\beta_i$ neurons with the lowest $MBS$ to add into $\mathbb{A}$.
11:     **end for**
12: **end for**

---

stage. Following the first principle, we first define BS to be the differential loss format.

$$BS = \Delta\mathcal{L}_{D_c} = \mathcal{L}_{D_c}(w_0) - \mathcal{L}_{D_c}(w), \tag{1}$$

where $w$ denotes the parameters of neurons in a trained (backdoored) network $\mathcal{F}'$ and $w_0$ is the initial parameters that usually approximates to 0 (Sun, 2019). Then, combined with the second principle, we can rewrite the computation of Eq. 1 through Taylor expansion.

$$BS \approx \mathcal{L}_{D_c}(0) - \mathcal{L}_{D_c}(w) = (\mathcal{L}_{D_c}(w) + \nabla_w\mathcal{L}_{D_c}(w)^T(0 - w) + R_1) - \mathcal{L}_{D_c}(w)$$
$$= -\nabla_w\mathcal{L}_{D_c}(w)^T w + R_1 \approx -\nabla_w\mathcal{L}_{D_c}(w)^T w, \tag{2}$$

where $R_1$ is the Taylor remainder consisting of 2-th derivative of $F_{D_c}(w)$ and quadratic term of $w$, which can be omitted in the final deduction. Note that, if $L_{D_c}(\cdot)$ is a non-linear function, the errors induced by the ignorance of $R_1$ can be non-negligible. However, for most neural network architectures, the last a few layers (output layer) are always fully connected, i.e., being *linear* layers. At this time, the linear approximation in Eq. 2 becomes reasonable and effective. Therefore, in practice, WIPER pays more attention to purify the top fully-connected layers with BS to achieve backdoor defense. We argue that such a setting is effective due to the following reason. Consider the scenario where the attacker only backdoors the feature extraction layers. The extracted feature about triggers can be simply blocked by the output layer if the neurons related to the trigger features are allocated with low weights. Also, the blocking can easily happen because the trigger features are useless for normal samples. Conversely, if the output layer is backdoored, the attack can be always successful as soon as the trigger feature is extracted. Therefore, to ensure attack effectiveness, the objective of most backdoor attacks is focused more on the output layers. The same goes for the defender. The experimental results in Section 5.5 also prove the above statement.

Moreover, Eq. 2 indicates that BS can be used to quantify the importance of $w$. Then, the importance of a certain neuron can be quantitatively evaluated by the summation of BSs of its all downstream weights. Intuitively, higher BS indicates higher contribution of the neuron to the loss decreasing of $\mathcal{F}'$, i.e., with more importance. Therefore, as illustrated in Algorithm 1 (Step 4), WIPER marks neurons with the lowest BS as bad neurons, and the downstream weights of the marked neurons are punished to diminish the impact of the neurons. Moreover, reconsider the backward propagation based gradient computation process. It can be observed that BS not only considers the activation weight of each neuron but also involves its connection with downstream neurons into importance evaluation via the first-order gradient $\nabla_w\mathcal{L}_{D_c}(w)^T$. Thus, unlike the conventional metric AM[1], BS can better avoid the problem of over-pruning good neurons.

---

[1] In prior backdoor defense methods (Liu et al., 2018), AM is computed by $AM = |w \cdot x|$ where $w$ is the neuron parameters and $x$ in the input.

Except for BS, we define, that can be adopted to evaluate the importance of a neuron, there are many inspiring metrics that own the similar function, a typical one of which is neuron shapley (Ghorbani & Zou, 2020). Theoretically, neuron shapley can perfectly assess the contribution of a neuron to the overall performance of the model but seriously suffers from the unaffordable exponential computation costs (even if with some mitigation measurements such as pruning algorithms or Monte Carlo estimation (Ghorbani & Zou, 2020)). For backdoor defense, it requires not only the excellent ability of backdoor erasing but also acceptable time complexity in applications. Therefore, we argue that neuron shapley is not a suitable metric to evaluate the importance of a neuron in backdoor defense scenarios. Thus, at the beginning of our design, we decided to abandon these complex methods and propose a new metric BS to resolve the problem.

### 4.1.2 FURTHER DISCUSSION

While implementing WIPER, we notice that although WIPER converges in the expected direction, its convergence curve suffers from great oscillation. Review the definition of BS in Eq. 2. It can be discovered that the unstability is mainly caused by the fact that limited by the real-world computation resource, WIPER sometimes has to adopt Stochastic Gradient Descent (SGD) to complete model purifying. With SGD, only one batch of data is taken into computation thereby easily leading the gradients used to compute BSs to be trapped into local optimum thereby degrading the performance of WIPER. At this time, we can use the momentum accumulation mechanism to mitigate the problem as follows.

$$MBS_i(w) = \eta \cdot MBS_{i-1}(w) + (1 - \eta) \cdot BS(w), 0 \leq \eta \leq 1, \tag{3}$$

where $MBS_i(w)$ denotes the accumulated BS of $w$ in the $i$-th epoch and $\eta$ is the momentum factor given by defenders.

Furthermore, in WIPER, the initial proportion $\beta$ of purified neurons (with least BS) is an empirical hyperparameter. Considering that the neurons required to be purified will be on the decrease along with the model purifying process, $\beta$ usually starts with a large value and then is slowly decayed until the purifying process is completed. Referring to the decaying mechanism of learning rate, we define the following decaying function in WIPER.

$$\beta = \beta_0 \cdot \frac{epoch - cur\_epoch}{epoch}, \tag{4}$$

where $\beta_0$ is initial value of $\beta$, $epoch$ is the number of training epochs and $cur\_epoch$ denotes the current epoch.

### 4.2 NEURON PURIFYING

We now elaborate how WIPER leverages an improved neuron fine-tuning strategy to achieve bad neuron purifying.

Without consideration of model performance, a basic idea for neuron purifying is to wipe out bad neurons by forcing all the parameters of them to be $0$. Following the idea and adding the model performance into consideration, we can derive the following way to achieve bad neuron purifying like (Liu et al., 2018).

$$\mathcal{L}_{purifying} = \sum_{i=1}^{n} CE(\mathcal{F}'(x_i), y_i) + \alpha \sum_{w \in \mathbb{A}} ||w - 0||_k, \tag{5}$$

where $\mathbb{A}$ denotes the set of bad neurons, $CE(\cdot)$ is the cross-entropy loss function, $\alpha$ is the penalty coefficient and $||w - 0||_k$ is the $L_k$-norm regularization term ($k \in \{1, 2\}$) that promotes the parameters of bad neurons to lose impact on the model outputs. Compared to $L_2$-regularization, $L_1$-regularization is known to be able to completely cleanse the impact of $w$ but requires more optimization rounds as $w$ is large (Hoefler et al., 2021). In contrast, $L_2$-regularization can accelerate the purifying process but can never makes $w$ to be exactly $0$ thereby easily causing the overshoot of the global optimum (also discussed in Section 5.6). To address the issue, we design a piece-wise regularization item called adaptive regularization (AR).

$$AR(w) = \begin{cases} -e^{-w-1}, & w < -1, \\ |w|, & |w| \leq 1, \\ e^{w-1}, & w > 1. \end{cases} \tag{6}$$

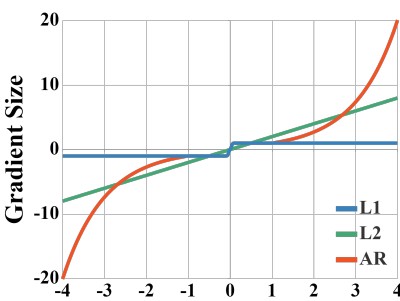

Figure 2: The penalty degree of three types of regularization items over different magnitudes of inputs

As shown in Fig. 2, AR can provide a higher penalty degree than $L_2$ to accelerate convergence when the input parameter is large. Meanwhile, the penalty degree of AR is similar to the $L_1$-norm regularization as the parameter is close to 0. In our evaluation, we will demonstrate that benefited from the ability to process different magnitudes of parameters, AR ensures faster convergence rate and high stability than $L_1$ and $L_2$ regularization in neuron fine-tuning (see Fig. 8 in Section 5.6). Finally, with AR, the purifying loss can be rewritten as follows:

$$\mathcal{L}_{purifying} = \sum_{i=1}^{n} CE(\mathcal{F}'(x_i), y_i) + \alpha \sum_{j \in \mathbb{A}} AR(w_j). \quad (7)$$

## 5 EXPERIMENTS

In this section, we mainly experiment with the effectiveness of WIPER in backdoor defense and answer the following research questions (**RQs**).

- **RQ1 Effectiveness:** Can WIPER defend the state-of-the-art backdoor attacks?
- **RQ2 A Closer Look at BS:** Is BS a good metric in identifying bad neurons?
- **RQ3 Improvement:** Is AR more suitable for bad neuron purifying than the common regularization?
- **RQ4: Approximation Availability:** Is purifying output layers enough to defend backdoor attacks?

Table 1: The performance of WIPER compared with six state-of-the-art defense methods over ten backdoor attacks in CIFAR-10.

| Defense | Before | | Fine-tuning | | KD | | NAD | | Fine-Pruning | | IBAU | | ANP | | WIPER | |
|---|---|---|---|---|---|---|---|---|---|---|---|---|---|---|---|---|
| Attack | ACC | ASR | ACC | ASR | ACC | ASR | ACC | ASR | ACC | ASR | ACC | ASR | ACC | ASR | ACC | ASR |
| BadNet | 87.42 | 99.5 | 78.42 | 23.31 | 50.17 | 5.46 | 70.54 | **4.34** | 80.68 | 14.70 | 81.77 | 11.70 | 83.02 | 5.45 | **83.20** | 5.03 |
| BA | 87.67 | 99.77 | 78.69 | 18.10 | 50.66 | 6.94 | 70.58 | 13.51 | 80.77 | 25.06 | 80.97 | 6.14 | 82.26 | 5.46 | **82.82** | **5.22** |
| ETA | 87.2 | 97.24 | 75.57 | 28.62 | 50.93 | 9.71 | 71.68 | 10.42 | 81.85 | 15.79 | **82.85** | 9.03 | 81.20 | 6.45 | 82.41 | **5.80** |
| IA | 86.83 | 98.67 | 78.38 | 28.10 | 49.10 | 7.29 | 70.71 | 5.87 | 78.73 | 42.40 | **82.46** | 10.51 | 80.91 | 5.78 | 82.04 | **6.02** |
| SIG | 85.14 | 91.1 | 75.82 | 16.54 | 51.84 | 6.56 | 68.71 | **3.62** | 80.69 | 16.03 | 80.54 | 8.42 | 82.30 | 5.28 | **82.41** | 3.73 |
| TrojanNN | 86.83 | 98.67 | 79.46 | 73.37 | 51.27 | 9.23 | 71.71 | 34.98 | 80.40 | 67.92 | 82.63 | 12.24 | **84.80** | 6.79 | 84.67 | **4.68** |
| IMC | 87.10 | 99.78 | 78.40 | 77.56 | 50.62 | 8.66 | 70.86 | 33.73 | 81.62 | 75.63 | 44.54 | 15.87 | 80.04 | 7.43 | **80.92** | 6.77 |
| WaveNet | 85.87 | 97.68 | 77.24 | 18.10 | 51.87 | 7.55 | 70.41 | 13.35 | 81.45 | 17.94 | 80.66 | 9.96 | 82.48 | 7.21 | **82.59** | **5.81** |
| SSA | 86.30 | 98.21 | 77.52 | 18.30 | 51.41 | 9.56 | 71.38 | 14.32 | 82.63 | 16.49 | 78.82 | 8.36 | 80.48 | 6.54 | **81.70** | 4.76 |
| LogitAttack | 86.92 | 99.98 | 79.08 | 23.92 | 49.51 | 5.66 | 69.81 | 6.73 | 80.43 | 13.96 | 81.86 | 10.85 | **83.09** | 7.65 | 83.03 | **5.54** |

### 5.1 EXPERIMENT SETTINGS

**Baselines.** Four state-of-the-art defense methods are considered as baselines, which are Fine-Pruning (Liu et al., 2018), Fine-tuning (Truong et al., 2020), KD (Yoshida & Fujino, 2020), NAD (Li et al., 2021a), IBAU (Zeng et al., 2022), and ANP (Wu & Wang, 2021). We compare their defense effect with WIPER against ten backdoor attacks, namely badNet (Gu et al., 2017), blend attack (BA) (Chen et al., 2017), enhanced trigger attack (ETA) (Li et al., 2020b), invisible attack (IA) (Zhong et al., 2020), sinusoidal signal attack (SIG) (Barni et al., 2019), TrojanNN (Liu et al., 2017), IMC (Pang et al., 2020), WaveNet (Nguyen & Tran, 2021), SampleSpecifiedAttack (SSA) (Li et al., 2021b), and LogitAttack (Zhang et al., 2022) on three benchmark datasets SVHN, CIFAR-10, and CIFAR-100. For fair comparisons, all the configurations of these attacks and defense follow their corresponding original papers (Appendix A.3 and A.4). Moreover, all defense methods are implemented with access to parts of the clean data randomly selected from the training set (5% to the size of training sets but never used in the training process like (Yoshida & Fujino, 2020; Li et al., 2021a; Liu et al., 2018; Zeng et al., 2022)).

| Attack | BadNet | | BA | | ETA | | IA | | SIG | | TrojanNN | |
|---|---|---|---|---|---|---|---|---|---|---|---|---|
| Method | ACC | ASR | ACC | ASR | ACC | ASR | ACC | ASR | ACC | ASR | ACC | ASR |
| Before | 87.42 | 99.50 | 87.67 | 99.77 | 87.20 | 97.24 | 86.83 | 98.67 | 85.14 | 91.10 | 86.83 | 98.67 |
| Fine-Pruning + AM | 80.68 | 14.70 | 80.77 | 25.06 | 81.85 | 15.79 | 78.73 | 42.40 | 80.69 | 16.03 | 80.40 | 67.92 |
| Fine-Pruning + BS | 81.11 | 10.47 | 81.70 | 14.22 | 82.34 | 10.72 | 80.20 | 18.11 | 80.79 | 11.71 | 81.85 | 16.50 |
| Ours + AM | 81.72 | 12.35 | 82.03 | 17.28 | 81.97 | 12.78 | 81.51 | 30.55 | 81.88 | 15.17 | 83.09 | 55.32 |
| Ours + BS | **83.20** | **5.03** | **82.82** | **5.22** | **82.41** | **5.80** | **82.04** | **6.02** | **82.41** | **3.73** | **84.67** | **4.68** |

Table 2: Comparison of backdoor defense with different purifying strategies and neuron importance evaluation metrics on CIFAR10.

**Evaluation Metrics.** The performance of backdoor defense is evaluated with two widely-used metrics: the accuracy of the model on clean samples (ACC), and the attack success rate (ASR). ASR denotes the ratio of triggered samples that succeed in misleading the target model to output attacker-chosen predictions. An effective defense method should significantly lower ASR and remain high ACC. All metrics are average values computed with the whole testing set that are never used in the training or purifying stage.

**Other Setups.** While implementing WIPER, we fix $\alpha, \beta_0, \eta$ to be 0.01, 0.5, and 0.9. The number of epochs for neuron purifying is set to 10 similar to (Li et al., 2021a). The purified layer is the output layer (usually the last full connection layer in the network) (Liu et al., 2018). Moreover, we adopt standard SGD with a constant learning rate of 0.01 and batch size of 128. All experiments are conducted using ResNet18 as the target model with a workstation equipped with Nvidia V100. Parts of the results with SVHN and CIFAR-100 are left in the *supplementary material*.

## 5.2 RQ1: EFFECTIVENESS

To validate the effectiveness of WIPER, we first test and analyse the defense effect of WIPER against different backdoor attacks. Then, we compare the performance of WIPER with other defense methods. Table 1, Table 7 (Appendix A.2) and Table 8 (Appendix A.2) summarize the experimental result of five different backdoor defense approaches against six state-of-the-art backdoor attacks with SVHN, CIFAR-10, and CIFAR-100. In the tables, *Before* means the ACC and ASR of backdoored models before being purified.

Overall, it can be observed that the performance (simultaneously consider both ACC and ASR) of WIPER considerably outperforms the other backdoor defense methods under different settings. Specifically, considering the decrease of ASR, two state-of-the-art methods, KD and NAD, can achieve competitive performance compared with WIPER but notably suffers from the problem of model performance degradation. For example, to maintain the ASR to be less than 10%, KD sacrifices at most 35% loss of ACC while WIPER only 5%. For NAD, it attains similar performance to WIPER on most backdoor attacks but fails to defend the TrojanNN attack. This is because besides adding poisoned data into the training set, TrojanNN allows the attacker to select and manipulate specific neurons to enhance the memory of backdoored model about the trigger patterns. The model distillation method used in NAD make the purified model inherit the "enhanced memory" from the backdoored model thereby leading to the low defense effect of NAD on TrojanNN. In contrast, the bad neuron filtering mechanism of WIPER can accurately mark these backdoored neurons and lower the ASR of TrojanNN significantly. Furthermore, considering performance maintenance, WIPER can ensure lower ACC degradation of the model compared with other defense methods in almost all conditions. For instance, in SVHN (Table 7), with only less than 1% degradation of the model performance, WIPER can drop the ASR of six attacks to less than 5%. However, the other method still struggles against the attacks, especially for Fine-Pruning.

## 5.3 RQ2: A CLOSER LOOK AT BS

As mentioned before, one of the key factors to make WIPER outperform other methods is the introduction of BS. Here, comprehensive experiments are conducted to explain why BS can achieve such an improvement on backdoor defense from three perspectives: 1) statistical analysis; 2) attention analysis; 3) effect on neuron purifying. In the experiments, a ResNet18 network is trained with CIFAR10. During the training process, we backdoor the model according to the widely-used bench-

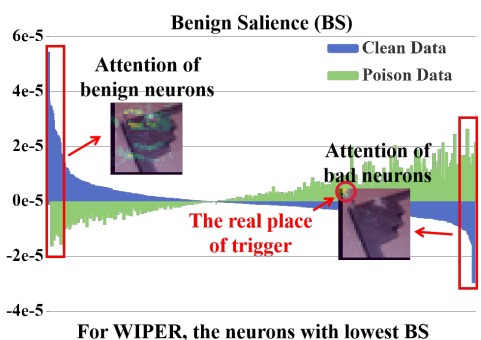
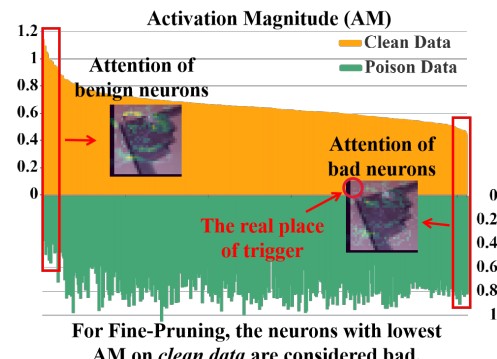

Figure 3: The left graph is about experimental result of BS, and the right one is about Fine-Pruning with AM. For both of the two graphs, the horizontal axis indicates the neurons of the victim model. Each bar corresponds to one neuron. All recorded BS and AM values are averaged over 1000 testing samples (500 for clean and 500 for poison data). Especially, the BS and AM for clean data are sorted in descending order for the convenience of observation.

mark attack BadNet (Gu et al., 2017) and ensure 100% ASR. The poison label and poison ratio are set to 0 and 5%. The trigger is a 3×3 square in the upper left corner (Gu et al., 2017).

**Statistical Analysis.** We record the averaged BSs of all neurons by feeding a batch of clean and poison data, 500 for each, as shown in the left of Figure 3. From the results, an interesting finding is that the BSs of neurons are almost opposite for clean and poison data. In fact, such a phenomenon is precisely what we desire in bad neuron filtering. This is because the phenomenon proves that BS well describes the principle of backdoor attacks: making some neurons to be strongly activated by triggers to change the original label output and rarely activated by clean data to avoid lowering the performance of victim model on normal inputs. Also, we can use BS to accurately mark the neurons with low importance to normal predictions but high significance to trigger activation. Referring to AM, shown in the right of Figure 3, it can be observed that there is no significant difference in AM between neurons. Such a flaw makes AM unable to provide a clear division between benign and bad neurons, and leads the bad neurons marked by AM to be mixed in some benign neurons.

**Attention Analysis.** Further, to validate the above statement, we visualize the attention of neurons on the original input image in Figure 3. The results show that for both BS and AM, the attention of benign neurons is mostly focused on the main body of the input image, i.e.,around the airplane. Then, the upper left corner, the place where triggers are added, draws the attention of bad neurons marked by BS. Clearly, it is proved that with BS, WIPER accurately identifies which neurons contribute the most to backdoor attacks. Conversely, the attention of the bad neurons marked by AM is not only focused on the upper left corner but also somewhere else.

**Effect on Neuron Purifying.** Finally, we conduct experiments to show that the choice of neuron importance evaluation metric can lead to significant effect on neuron purifying. From the results of Table 2, given the same purifying method (Fine-Pruning or WIPER), the defense effect with BS outperforms the ones with AM on both ACC and ASR. Especially, with AM, Fine-Pruning fails to defend the TrojanNN attack. However, with BS, Fine-Pruning succeeds in defending TrojanNN due to the better ability of BS to identify bad neurons. All the facts prove that BS is a good metric.

## 5.4 RQ3: IMPROVEMENT OF AR ON NEURON PURIFYING

To show the advantages of AR on fine-tuning based neuron purifying, we experimentally compare it with the other two commonly used regularization methods $L_1$ and $L_2$. Figure 8 (Appendix A.2) illustrates the performance of WIPER employed with three kinds of regularization mechanisms.

It is observed that the difference of these three kinds of regularization methods mainly stems from the changing trend of ASR. In more details, the intrinsic property of $L_2$ makes it only "strong" on penalizing large parameters (more than 1, shown in Figure 2), and unable to actually decrease parameters to 0. However, for a backdoored network, the parameters of bad neurons are usually

very small (close to 0). Thus, $L_2$ fails to provide proper penalty on bad neurons and leads to the lowest ASR decreasing speed. Then, for $L_1$, it can always reduce the ASR to a similar level to AR . However, since the penalty of $L_1$ is too weak, its convergence speed is still lower than AR. Finally, the required epochs of AR is just around 2, which is significantly lower than $L_1$ and $L_2$ regularization, which requires more than 15 epochs to erase the backdoor. Thus, the adoption of AR in backdoor defense can greatly improve the performance of neuron purifying.

## 5.5 RQ4: Is purifying output layers enough to defend backdoor attacks

In this subsection, we mainly answer the question of whether only purifying the last few layers is enough for erasing the backdoor from the victim model. Table 5 (Appendix A.2) reports the performance of WIPER as the target purifying layers are changed. From the results, adding feature extraction layer (i.e., convolution layers) into backdoor purifying shows little improvement on defense effect, only less 1% more decrease on ASR. In contrast, the performance of the purified model degrades more significantly when more convolution layers are introduced. As aforementioned, such a phenomenon occurs because backdoor attacks usually pay less attention on the feature extraction layers than the output layer to achieve better attack effectiveness. Therefore, forcibly purifying the feature extraction layers contributes little to the backdoor defense but can easily lead to the over-pruning of the victim model and lower the model performance.

## 5.6 Other Cases

**Impact of Clean Data.** As discussed in the prior works (Liu et al., 2018; Truong et al., 2020; Yoshida & Fujino, 2020; Li et al., 2021a), the amount of clean data owned by the defender is a crucial factor that affects the effect of backdoor defense significantly. Here, for the convenience of statistic and comparison, we use the ratio of clean data set to training set as the indicator to denote the amount of clean data owned by the defender. Figure 5 (Appendix A.2) demonstrates the performance of five defense methods against six attacks over different data ratios. Strikingly, with the strict condition of 1% data ratio, WIPER can still maintain almost 90% ASR drops and less than 3.5% accuracy drops. In contrast, most of other methods fail to achieve effective backdoor defense (less than 40% ASR drops on some attacks) or suffer from significant ACC drops (more than 20%).

**Impact of $\alpha$.** As discussed in Section 4.2, $\alpha$ is a vital factor to determine the degree of WIPER to punish the bad neurons. Table 6 (Appendix A.2) summarizes the experimental results of WIPER with coarse-tuning of $\alpha$. The results show that larger $\alpha$ makes WIPER focus more on backdoor purifying and achieve lower ASR but is with the expense of slightly decreased ACC. Correspondingly, lower $\alpha$ allows the target model to maintain higher ACC but results in a higher ASR. Therefore, a trade-off exists between ASR and ACC. However, the trade between ASR and ACC are not exchanged at equal values. It can be found that in most cases, only less than 1% ACC drop can leads to the drop of ASR for about 50% to 60%. As a result, in practice, the defender is suggested to choose a litter higher magnitude of $\alpha$ to ensure the effectiveness of WIPER.

**Impact of Purifying Strategy.** Furthermore, in Table 2, we also show the influence of purifying strategy choice on backdoor defense. As expected, despite of maintaining the performance of the purified model in an acceptable range, Fine-Pruning (fine-tuning with $L_1$ + pruning) still suffers from poor capacity of backdoor erasing. However, if Fine-Pruning is replaced with our purifying strategy, the results can be improved significantly. For instance, referring to the TrojanNN attack, substituting Fine-Pruning with our purifying strategy can make the ACC rise from 81.85% to 84.67% while the ASR drops from 16.50% to 4.68%, even with AM as the importance evaluation metric.

## 6 Conclusion

We proposed a backdoor defense approach WIPER that could combine the advantages of both model pruning and fine-tuning. WIPER improved current backdoor defense method from two aspects. The first was to propose a new metric called benign salience to evaluate the importance of neurons in a network and filter the bad ones. The second was to design a new regularization item for backdoor purifying, which could accelerate the neuron purifying process significantly. Finally, we conducted extensive experiments to show that the WIPER achieved ideal performance on both erasing the backdoor and maintaining the model performance after being equipped with such two techniques.

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

# A APPENDIX

## A.1 ALTERNATIVE METRICS

Here we first examine whether the gradient magnitude of a neuron can be served as a good metric to distinguish between good and bad neurons and Figure 4 shows results. Similar to AM, solely employing gradient magnitude is hard to detect bad neurons.

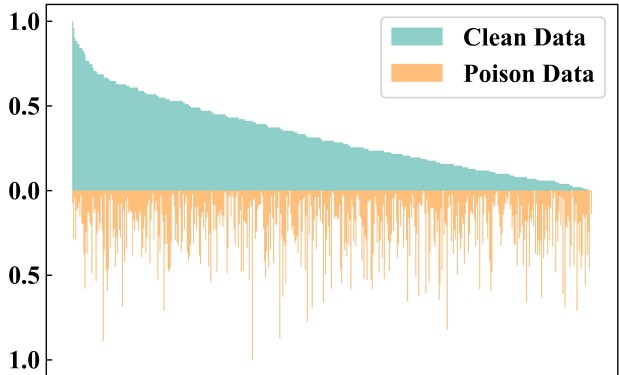

Figure 4: The gradient magnitude of each neuron with respect to clean data and poison data respectively. The evaluation settings follow Figure 3.

**BS vs. Neuron Shapley & Integrated Gradient.** Some literature (Sundararajan et al., 2017; Ghorbani & Zou, 2020) developed various metrics to quantify the importance of neurons. To show better practicality of BS, we compare our method with other importance evaluation metrics. Specifically, we select two common-used metrics: Neuron Shapley (Ghorbani & Zou, 2020) and Integrated Gradient Sundararajan et al. (2017). Table 3 reports the performance and overhead of three metrics. As can be seen, BS obtains competitive defense performance and only needs considerably smaller overheads compared with Integrated Gradient and Shapley value.

Table 3: The performance and overhead of different metrics against BadNet.

| Metric | ACC | ASR | Time |
|---|---|---|---|
| BS | 83.2 | 5.03 | 1.00 |
| Integrated Gradient | 83.25 | 5.10 | 10.76 |
| Shapley Value | 82.89 | 4.92 | 968.17 |

Table 4: The performance of AI-Lancet and WIPER against four backdoor attacks.

| Defense | AI-Lancet | | WIPER | |
|---|---|---|---|---|
| Attack | ACC | ASR | ACC | ASR |
| BadNet | 80.15 | 5.21 | **83.20** | **5.03** |
| BA | 73.81 | 27.17 | **82.82** | **5.22** |
| ETA | 68.67 | 29.93 | **82.41** | **5.80** |
| SSA | 68.47 | 22.14 | **81.7** | **4.76** |

## A.2 ADDITIONAL EXPERIMENT ON DIFFERENT DATASETS

Firstly, we demonstrate that the poor defense performance of AI-Lancet Zhao et al. (2021) against state-of-the-art attacks. Table 4 shows that AI-Lancet is vulnerable against BA, ETA, and SSA, which adopt more threatening triggers covering the entire images or being dynamic.

| Attack | BadNet | | BA | | ETA | | IA | | SIG | | TrojanNN | |
|---|---|---|---|---|---|---|---|---|---|---|---|---|
| Purified Layer | ACC | ASR | ACC | ASR | ACC | ASR | ACC | ASR | ACC | ASR | ACC | ASR |
| FC only | **83.20** | 5.03 | **82.82** | 5.22 | **82.41** | 5.80 | **82.04** | 6.02 | **82.41** | 3.73 | **84.67** | 4.68 |
| FC + 1 Conv | 82.19 | 5.15 | 81.29 | 5.93 | 81.37 | 5.77 | 80.46 | 5.44 | 81.26 | 3.98 | 82.89 | 4.24 |
| FC + 2 Conv | 80.29 | 4.79 | 79.31 | 5.24 | 79.42 | 5.38 | 77.66 | 5.69 | 79.82 | 4.05 | 81.00 | **4.01** |
| FC + 3 Conv | 77.03 | 4.83 | 77.84 | **4.34** | 77.65 | 4.60 | 76.71 | 5.62 | 77.96 | 3.45 | 79.72 | 5.36 |
| FC + 4 Conv | 74.49 | **4.51** | 74.86 | 4.35 | 71.56 | **4.04** | 73.33 | **4.96** | 74.56 | **3.22** | 72.73 | 4.67 |

Table 5: The performance of WIPER when purifying different layers. Here $n$ Conv indicates purifying the last n convolution layers.

| Attack | BadNet | | BA | | ETA | | IA | | SIG | | TrojanNN | |
|---|---|---|---|---|---|---|---|---|---|---|---|---|
| $\alpha$ | ACC | ASR | ACC | ASR | ACC | ASR | ACC | ASR | ACC | ASR | ACC | ASR |
| 0.001 | **83.57** | 77.03 | **84.09** | 86.74 | 83.69 | 66.43 | **84.07** | 76.82 | **83.57** | 28.09 | 86.09 | 33.18 |
| 0.005 | 83.55 | 21.01 | 84.14 | 26.20 | **84.41** | 16.54 | 83.44 | 25.97 | 82.64 | 6.69 | **86.22** | 5.57 |
| 0.01 | 83.20 | **5.03** | 82.82 | 5.22 | 82.41 | 5.80 | 82.04 | 6.02 | 82.41 | 3.73 | 84.67 | 4.68 |
| 0.05 | 82.76 | 5.48 | 82.89 | 5.42 | 82.10 | 5.48 | 81.97 | 5.69 | 81.78 | 4.02 | 84.25 | 4.86 |
| 0.1 | 82.20 | 5.30 | 82.18 | **5.07** | 81.55 | **3.96** | 81.33 | **4.72** | 81.20 | **3.44** | 84.39 | **4.39** |

Table 6: The performance of WIPER with different $\alpha$ against six backdoor attacks in CIFAR-10.

Here we report some additional experimental results in SVHN and CIFAR-100. From experiments, we can further consolidate our conclusion that WIPER is an effective defense against backdoor attacks (also discussed in Section 5).

Figure 6 and Figure 7 illusrate the defense performance of different defenses on SVHN and CIFAR-100. Besides, Figure 9 and Figure 10 show the defense performance of WIPER with varying regularization items on SVHN and CIFAR-100. Table 9 and Table 10 report the defense performance of WIPER with different purifying strategies and neuron importance evaluation metrics on SVHN and CIFAR-100.

| Defense | Before | | Fine-tuning | | KD | | NAD | | Fine-Pruning | | IBAU | | WIPER | |
|---|---|---|---|---|---|---|---|---|---|---|---|---|---|---|
| Attack | ACC | ASR | ACC | ASR | ACC | ASR | ACC | ASR | ACC | ASR | ACC | ASR | ACC | ASR |
| BadNet | 94.68 | 100 | 85.74 | 3.17 | 60.16 | 4.42 | 86.22 | 2.71 | 93.17 | 69.57 | 89.70 | 6.83 | **94.14** | **0.49** |
| BA | 94.62 | 100 | 86.54 | 1.44 | 52.4 | 2.61 | 86.47 | 3.11 | 92.29 | 79.65 | 87.95 | 9.91 | **94.09** | **1.43** |
| ETA | 94.59 | 99.98 | 87.61 | 6.34 | 53.08 | 4.98 | 87.56 | 6.28 | 93.41 | 68.52 | 91.22 | 10.98 | **94.11** | **4.94** |
| IA | 94.68 | 100 | 85.17 | 3.26 | 52.45 | 4.51 | 86.25 | **1.87** | 94.23 | 87.53 | 89.71 | 9.45 | **94.53** | 2.48 |
| SIG | 94.38 | 96.97 | 84.85 | 1.31 | 53.22 | 6.87 | 85.3 | 3.22 | 93.16 | 38.53 | 88.59 | 2.28 | **93.69** | **1.31** |
| TrojanNN | 94.71 | 100 | 88.75 | 8.42 | 49.64 | 4.81 | 87.45 | 31.97 | 92.31 | 97.82 | 88.22 | 11.34 | **94.52** | **2.41** |
| IMC | 94.15 | 100 | 87.52 | 10.56 | 52.62 | 5.15 | 86.41 | 32.53 | 93.50 | 98.23 | 87.52 | 18.16 | **93.90** | **1.92** |
| WaveNet | 94.13 | 99.36 | 84.68 | 2.81 | 55.86 | 6.26 | 84.84 | 4.57 | 92.54 | 40.86 | 88.06 | 3.25 | **93.90** | **2.70** |
| SSA | 94.60 | 97.41 | 85.22 | 2.22 | 52.83 | 7.59 | 85.30 | 4.07 | 94.95 | 40.76 | 87.52 | 5.27 | **93.33** | **3.17** |
| LogitAttack | 94.93 | 100 | 86.16 | 3.38 | 51.68 | 4.22 | 86.30 | 3.55 | 93.60 | 68.75 | 90.01 | 6.62 | **93.51** | **2.14** |

Table 7: The performance of WIPER compared with four state-of-the-art defense methods over six backdoor attacks in SVHN.

| Defense | Before | | Fine-tuning | | KD | | NAD | | Fine-Pruning | | IBAU | | WIPER | |
|---|---|---|---|---|---|---|---|---|---|---|---|---|---|---|
| Attack | ACC | ASR | ACC | ASR | ACC | ASR | ACC | ASR | ACC | ASR | ACC | ASR | ACC | ASR |
| BadNet | 69.66 | 99.83 | 60.74 | 11.01 | 18.84 | 1.51 | 32.80 | 2.32 | 63.48 | 25.53 | 59.29 | 3.44 | **63.71** | **1.12** |
| BA | 68.68 | 99.81 | 61.68 | 23.32 | 15.32 | 1.59 | 33.24 | 2.46 | 63.08 | 26.68 | 61.62 | 9.08 | **63.21** | **0.80** |
| ETA | 68.98 | 98.43 | 62.71 | 17.99 | 17.52 | 1.44 | 31.03 | 3.11 | **62.89** | 37.63 | 60.82 | 1.67 | 62.35 | **0.83** |
| IA | 69.12 | 99.88 | 59.44 | 27.79 | 16.57 | 1.91 | 31.37 | 3.02 | 60.97 | 23.30 | 60.27 | 5.94 | **62.11** | **0.55** |
| SIG | 68.92 | 90.21 | 57.48 | 21.18 | 15.48 | 1.58 | 30.50 | 1.62 | 61.43 | 28.07 | 59.11 | 8.12 | **63.89** | **0.26** |
| TrojanNN | 70.84 | 94.66 | 61.97 | 56.76 | 16.61 | 1.57 | 30.38 | 32.03 | 61.45 | 78.99 | 60.58 | 11.56 | **63.92** | **0.20** |
| IMC | 69.08 | 99.55 | 62.41 | 62.15 | 17.52 | 1.50 | 31.19 | 35.40 | 61.62 | 87.98 | 60.47 | 15.16 | **60.91** | **0.97** |
| WaveNet | 67.68 | 98.84 | 59.49 | 22.07 | 16.00 | 1.29 | 31.08 | 2.46 | 62.04 | 27.87 | 59.95 | 8.37 | **62.04** | **0.99** |
| SSA | 68.03 | 97.76 | 58.80 | 22.00 | 15.50 | 1.38 | 30.45 | 0.21 | 62.23 | 28.98 | 61.00 | 6.46 | **60.82** | **1.38** |
| LogitAttack | 68.35 | 99.87 | 61.49 | 11.26 | 19.24 | 2.55 | 32.90 | 3.15 | 62.63 | 26.09 | 58.80 | 4.82 | **62.00** | **0.53** |

Table 8: The performance of WIPER compared with four state-of-the-art defense methods over six backdoor attacks in CIFAR-100.

## A.3 ATTACK IMPLEMENTATION DETAILS

We summarize the implementation details of ten backdoor attacks used in this paper as follows:

| Attack | BadNet | | BA | | ETA | | IA | | SIG | | TrojanNN | |
|---|---|---|---|---|---|---|---|---|---|---|---|---|
| Method | ACC | ASR | ACC | ASR | ACC | ASR | ACC | ASR | ACC | ASR | ACC | ASR |
| before | 94.68 | 100.00 | 94.62 | 100.00 | 94.59 | 99.98 | 94.68 | 100.00 | 94.38 | 96.97 | 94.71 | 100.00 |
| Pruning + AM | 93.17 | 69.57 | 92.29 | 79.65 | 93.41 | 68.52 | 94.53 | 87.53 | 93.16 | 38.53 | 92.31 | 97.82 |
| Pruning + BS | 93.58 | 17.58 | 93.01 | 19.25 | 93.58 | 24.92 | 93.07 | 13.78 | 93.52 | 12.84 | 93.41 | 20.88 |
| Purifying + AM | 93.70 | 34.06 | 93.91 | 29.32 | 93.56 | 45.50 | **94.57** | 35.19 | 93.64 | 27.10 | 93.84 | 63.08 |
| Purifying + BS | **94.14** | **0.49** | **94.09** | **2.87** | **94.11** | **4.94** | 94.23 | **9.48** | **93.69** | **1.48** | **94.52** | **2.41** |

Table 9: Comparison of backdoor defense with different purifying strategies and neuron importance evaluation metrics on SVHN.

| Attack | BadNet | | BA | | ETA | | IA | | SIG | | TrojanNN | |
|---|---|---|---|---|---|---|---|---|---|---|---|---|
| Method | ACC | ASR | ACC | ASR | ACC | ASR | ACC | ASR | ACC | ASR | ACC | ASR |
| before | 69.66 | 99.83 | 68.68 | 99.81 | 68.98 | 98.43 | 69.12 | 99.88 | 68.92 | 90.21 | 70.84 | 94.66 |
| Pruning + AM | 63.48 | 25.53 | 63.08 | 26.68 | 62.89 | 37.63 | 60.97 | 23.30 | 61.43 | 28.07 | 61.45 | 78.99 |
| Pruning + BS | 63.55 | 16.69 | 63.02 | 19.87 | 61.73 | 18.07 | 61.02 | 11.77 | 62.21 | 15.28 | 61.96 | 29.63 |
| Purifying + AM | 63.62 | 20.31 | 63.15 | 23.51 | **63.01** | 25.04 | 61.75 | 19.27 | 62.52 | 22.67 | 62.83 | 70.57 |
| Purifying + BS | **63.71** | **1.12** | **63.21** | **0.80** | 62.35 | **0.83** | **62.11** | **0.55** | **63.89** | **0.26** | **63.92** | **0.20** |

Table 10: Comparison of backdoor defense with different purifying strategies and neuron importance evaluation metrics on CIFAR100.

- **BadNet:** A 3×3 trigger with random pixel values is pasted in the top left corner of 5% training images, and labels tamper with 0.

- **Blend Attack:** We used the same trigger, a hello kitty image, in the original paper, blend ratio of 0.2, and inject rate of 0.05.

- **Enhanced Trigger Attack:** Following the original paper, we adopt the same random spatial transformation layer consisting of rotation and scale to pre-process the poison data. The associated parameters of the trigger used in this attack are identical to BadNet.

- **Invisible Attack:** We generated the same trigger with $32 \times 32$ resolution in the original paper and injection rate of 0.05.

- **Sinusoidal signal attack:** We used the backdoor trigger generation function in the original paper with $\delta = 20$ and $f = 6$ and injection rate of 0.1.

- **TrojanNN:** Based on the implementation of the original paper, we utilized the same reverse engineer technology to craft a $3 \times 3$ square trigger from the fully connected layer. Other parameters are set the same in BadNet.

- **IMC:** Similar to the original paper, we used a random noise with $3 \times 3$ size as the trigger and optimize the trigger during the training process.

- **WaveNet:** We used the corruption technique with a transformation probability of 0.2 defined in the original paper to process images.

- **SSA:** We added noises produced by the generator used in the original paper for 20% of images.

- **LogitAttack:** We poisoned 5% training data with the trigger of $3 \times 3$ similar to BadNet.

## A.4 DEFENSE IMPLEMENTATION DETAILS

We list the detailed defense settings of Fine-Pruning, Fine-tuning, KD, NAD, and I-BAU for reference:

- **Fine-Pruning:** As suggested in the original paper, we pruned the last layer of the model with a pruning rate of 0.1, where AM is measured over a random subset of the training set as same to BS.

- **Fine-tuning:** Following the original paper, we adopted a standard fine-tuning procedure with a fixed learning rate of 0.1, momentum factor of 0.9, $L_2$ weight decay factor of $1 \times 10^{-4}$, and cross-entropy loss function to recover the backdoored model.

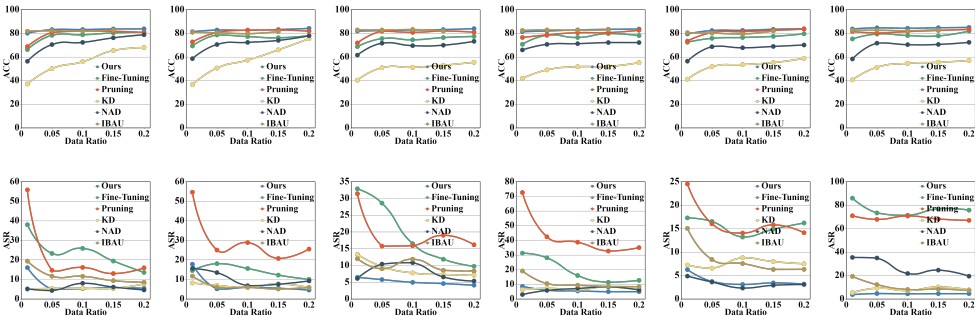

Figure 5: The defense performance of five defense methods over different data ratio (0.01, 0.05, 0.1, 0.15, 0.2) in CIFAR-10. The images from left to right indicate against BadNet, BA, ETA, IA, SIG, and TrojanNN.

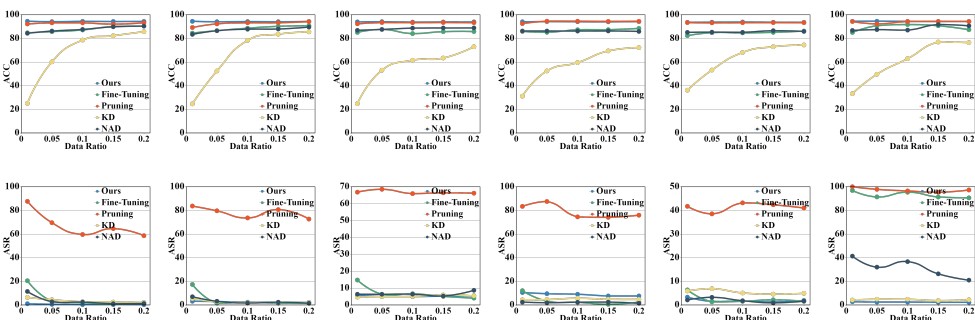

Figure 6: The defense performance of five defense methods over different data ratio (0.01, 0.05, 0.1, 0.15, 0.2) in SVHN.

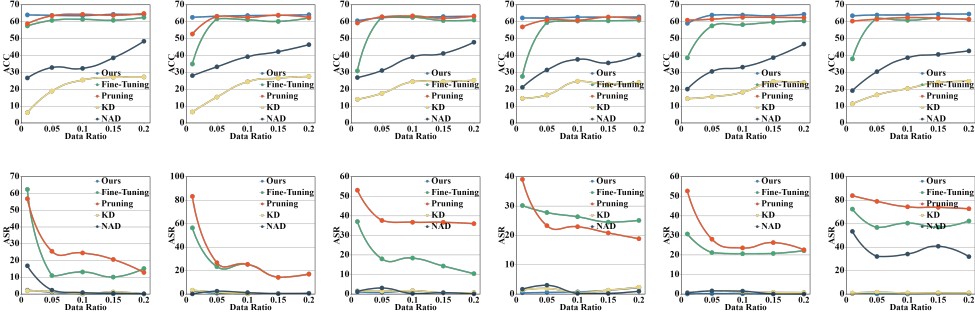

Figure 7: The defense performance of five defense methods over different data ratio (0.01, 0.05, 0.1, 0.15, 0.2) in CIFAR-100.

- **KD:** The standard knowledge approach is employed to recover the backdoored model, where cross-entropy loss and KL divergence are fused together as the overall loss function. The weights of the two loss terms are set to 0.5 along with the temperature of 2 and epoch of 50.

- **NAD:** We replicated the defense of NAD, where the backdoored model is fine-tuned for 10 epochs with an initial learning rate of 0.1 and a momentum of 0.9. The learning rate is divided by 10 after every 2 epochs.

- **I-BAU:** For I-BAU, we reused their original hyperparameter $C_\delta = 10$.

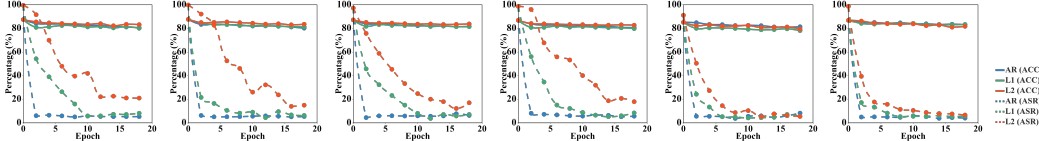

Figure 8: The performance of WIPER with different regularization items ($L_1$, $L_2$, and $AR$) on CIFAR-10. The images from left to right indicate against BadNet, BA, ETA, IA, SIG, and TrojanNN.

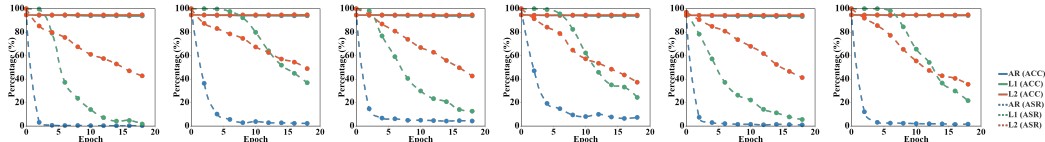

Figure 9: The performance of WIPER with different regularization items ($L_1$, $L_2$, and $AR$) on SVHN.

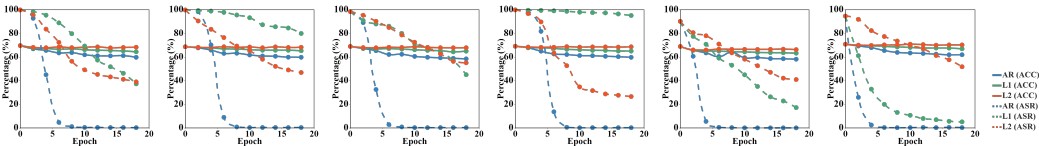

Figure 10: The performance of WIPER with different regularization items ($L_1$, $L_2$, and $AR$) on CIFAR-100.

