# OpenReview forum: "Defense against Backdoor Attacks via Identifying and Purifying Bad Neurons"
_ICLR.cc/2023/Conference — Submitted to ICLR 2023_

### Official Review · Reviewer_1xEU · 2022-10-21

**Confidence:** 3
**Correctness:** 3
**Technical Novelty And Significance:** 2
**Empirical Novelty And Significance:** 2
**Recommendation:** 5

**Clarity, Quality, Novelty And Reproducibility:**

I had clarity issues while reading the paper. A major line of work identifying important neurons is missing.

**Strength And Weaknesses:**

The results are promising.

**Summary Of The Paper:**

The paper presents a neuron-based method to defense against backdoor attacks. They first identify less important neurons in the network and then purify them using a fine-tuning based method. The paper contributes to both the selection of less important neurons and in purifying the network. Compared to the common method of selecting less important neurons based on the activation magnitude, authors argued that activation ignores the strong connections that a neuron may have and may result in selecting important neurons as the least important neurons. The authors proposed Benign salience (BS) that ranks the neurons based on how they contribute to reducing the loss. The neurons that contributed the least are identified as bad neurons. The bad neurons are then purified using an improved fine-tuning strategy. The experiments show that the proposed method outperforms all the baseline methods.

**Summary Of The Review:**

- I found the writing of the paper a bit problematic. It mentions WIPER as their method but then mainly presents BS as their method. I did not get whether WIPER refers to only improved fine-tuning strategy or it refers to BS plus improved fine-tuning strategy.

- The paper mainly ignores various methods to identify the role of a neuron with respect to the prediction such as integrated gradient, deep lift, layer-wise relevance back propagation. Looking at just activation is indeed not the best way to identify important neurons but I would expect authors to compare their proposed method with the methods from the literature.

- Figure 2 (left): The fine neurons that authors are getting could be the result of the BadNet method that it learns to cleanly separate the poison neurons. Testing gradient based methods on this model would clarify the benefit of BS in comparison to other methods.

- (Section 4.2) I did not fully get the description of L1 and L2. L1 brings sparsity so it should turn the neurons will low weight/activations to zero quickly and L2 normalizes the spikes in the scores and it won't turn neurons to zero. Section 5.4 seems to be correctly discussing the effect of L1 and L2.

---

> ### Author Response · Authors · 2022-11-16
> **Response (1/1)**
>
> Question 1: I found the writing of the paper a bit problematic. It mentions WIPER as their method but then mainly presents BS as their method. I did not get whether WIPER refers to only improved fine-tuning strategy or it refers to BS plus improved fine-tuning strategy.
>
> Response:
> WIPER refers to BS plus improved fine-tuning strategy.
> We have revised Section 4 to make our approach clearer.
>
>
> Question 2: The paper mainly ignores various methods to identify the role of a neuron with respect to the prediction such as integrated gradient, deep lift, layer-wise relevance back propagation. Looking at just activation is indeed not the best way to identify important neurons but I would expect authors to compare their proposed method with the methods from the literature.
>
> Response:
> Based on your suggestion, we examine the effectiveness of different metrics.
> In detail, Shapley value [1] and integrated gradient [2] are considered to defend against backdoor attacks (BadNet).
> Since we find that some metrics are expensive to evaluate, the following table also reports the defense performance together with running overhead.
> Overall, these metrics achieve similar defense performance but the baseline metrics consume 10.76x and 968.17x more time compared with BS.
>
> |        Metric       |  ACC  |  ASR |  Time  |
> |:-------------------:|:-----:|:----:|:------:|
> |          BS         |  83.2 | 5.03 |  1.00  |
> | Integrated Gradient | 83.25 | 5.10 |  10.76 |
> |    Shapley Value    | 82.89 | 4.92 | 968.17 |
>
>
> [1] Ghorbani, Amirata and James Y. Zou. “Neuron Shapley: Discovering the Responsible Neurons.” ArXiv abs/2002.09815 (2020): n. pag.
>
> [2] Sundararajan, Mukund et al. “Axiomatic Attribution for Deep Networks.” ArXiv abs/1703.01365 (2017): n. pag.
>
> Question 3: Figure 2 (left): The fine neurons that authors are getting could be the result of the BadNet method that it learns to cleanly separate the poison neurons. Testing gradient based methods on this model would clarify the benefit of BS in comparison to other methods.
>
> Response:
> We examine whether gradient-based methods can be served as a good metric to distinguish good and bad neurons and we find that, compared with BS, gradient-based methods perform badly.
> See in Appendix A.1.
>
> Question 4: (Section 4.2) I did not fully get the description of L1 and L2. L1 brings sparsity so it should turn the neurons will low weight/activations to zero quickly and L2 normalizes the spikes in the scores and it won't turn neurons to zero. Section 5.4 seems to be correctly discussing the effect of L1 and L2.
>
> Response:
> We have revised this part.
> See in the revised paper (4.2).

---

### Official Review · Reviewer_etm6 · 2022-10-23

**Confidence:** 3
**Correctness:** 3
**Technical Novelty And Significance:** 2
**Empirical Novelty And Significance:** 2
**Recommendation:** 5

**Clarity, Quality, Novelty And Reproducibility:**

I think overall the paper is interesting and is acceptably clear. The main ideas and goals are clearly stated, and the results are positive. In my opinion, the experimental evaluation of the proposed method that tries to answer the 4 research questions posed is one of the main assets of the paper. However, I'm hesitant about the novelty and technical and methodological contribution of the proposed algorithm.

In terms of reproducibility, I'd say that the work is reproducible based on the main manuscript, the supplementary materials and the source code provided.

Finally, there are several aspects related to the presentation/organization of the paper that could be improved:
- the writing of the paper (to avoid typos like "priro" -> "prior" or "decrease parameters to 0.." -> "decrease parameters to 0.")
- Table 4 is not referenced/commented in the body of the manuscript.
- there is no global figure to introduce the whole approach and problematic in a visual manner (including an illustration of a backdoor attack).
- the algorithm employed to describe the proposed paper is included in Supplementary Materials, and not in the main paper.
- captions of some figures are not particularly informative. For instance, in Figure 1.

**Strength And Weaknesses:**

Strengths:
- empirical evaluation and effectiveness, i.e. the proposed method yields better quantitative results than the included competitor methods.
- relevant application: the fact of developing/designing safer and more resilient neural models is always welcome.

Weaknesses:
- not sure if technical/methodological novelty of this work with respect to the state-of-the-art is enough.
- not sure if authors are comparing (quantitatively and qualitatively) with all relevant prior work.
- the organization, presentation and writing of the paper.

**Summary Of The Paper:**

The paper proposes a novel backdoor defense method (called WIPER) that includes: i) a new metric, called benign salience, to identify backdoored neurons; and ii) a new adaptive regularization mechanism to assist in purifying the identified bad neurons.

**Summary Of The Review:**

See "Clarity, Quality, Novelty And Reproducibility" Section of this review.

---

> ### Author Response · Authors · 2022-11-16
> **Response (1/1)**
>
> Question 1: not sure if technical/methodological novelty of this work with respect to the state-of-the-art is enough.
>
> Response:
> Thanks for your question.
> As commented, neuron purifying is not a very fresh idea because almost all backdoor erasing methods have to ``purify'' neurons to some extent.
> However, even for now, how to purify backdoor neurons more thoroughly with less performance loss has always been a challenge.
> Especially, the attacker has never stopped to develop more stealthy and powerful attacks to break the past purifying methods.
> Motivated by this, our work explores some new ways to enhance the purifying-based defense strategy.
> Also, our experiments validate that the newly proposed work is the best to address the above challenge among all existing works.
>
>
> Question 2: not sure if authors are comparing (quantitatively and qualitatively) with all relevant prior work.
>
> Response:
> Combined with the comments from other reviewers, more baselines and metrics are involved in the revised version.
> For now, 6 baselines, 10 attacks, and 4 metrics (AM, BS, integrated gradient, and Shapley Value) are reported in our experiment (Table 2 and Table 3), which cover all the commonly used metrics and most of the outstanding works about backdoor attacks/defenses in the past years.
> Table 3 is in Appendix A.1.
>
>
> Question 3: the writing of the paper (to avoid typos like "priro" -> "prior" or "decrease parameters to 0.." -> "decrease parameters to 0.")
>
> Response:
> We have revised the paper to fix these errors.
>
> Question 4: Table 4 is not referenced/commented in the body of the manuscript.
>
> Response:
> We have revised the issue.
>
> Question 5: there is no global figure to introduce the whole approach and problematic in a visual manner (including an illustration of a backdoor attack).
>
> Response:
> We have added a global figure to show backdoor attacks and the overall idea of our metohd.
> Please see the revised paper (Figure 1).
>
> Question 6: the algorithm employed to describe the proposed paper is included in Supplementary Materials, and not in the main paper.
>
> Response:
> We have moved the algorithm into the main paper.

---

### Official Review · Reviewer_krJA · 2022-10-24

**Confidence:** 3
**Correctness:** 2
**Technical Novelty And Significance:** 2
**Empirical Novelty And Significance:** 2
**Recommendation:** 3

**Clarity, Quality, Novelty And Reproducibility:**

This paper misses many details to help readers to understand. Thus, it requires a major revision.


**Strength And Weaknesses:**

**Strength**
1. The empirical results demonstrate the effectiveness of the proposed method.
2. If I understand correctly, the author defines the importance of a weight parameter using the element-wise product of the weights and their gradients. There is a similar formulation (element-wise product of the input and the gradient) in Saliency Maps [1, 2]. If we could find the similarities between them, it must be very interesting.

**Weakness**
1. This paper is written badly and requires major revision. I have a terrible time understanding this paper. For example, when citing related works, please use brackets “(” & “)” in most cases to include them, i.e., use \citep in Latex.

2. It is difficult to understand the formulation of Eqn (2).
- Since DNN has an extremely large number of parameters, close-to-zero initialization still result in different loss compared to zero-initialization. Why could we replace  $L_{D_c}(w_0)$ with  $L_{D_c}(0)$?
- In this formulation, BS is defined for the whole model and every neuron share this value inside the model. So how can we distinguish neurons using this shared BS? Is the neuron-wise BS defined as the sum of the products of the parameters and their derivatives inside a specific neuron?

3. Where is Figure 4 in Sec 5.4?

4. If this paper regards it as a pruning method, it neglects an important work, ANP [3]. Since this is another pruning-based method, this paper should compare it.

[1] Avanti Shrikumar, Peyton Greenside, Anna Shcherbina, and Anshul Kundaje. Not just a black box: Learning important features through propagating activation differences. arXiv preprint arXiv:1605.01713, 2016.

[2] Julius Adebayo, Justin Gilmer, Michael Muelly, Ian Goodfellow, Moritz Hardt, Been Kim. Sanity Checks for Saliency Maps. In NeurIPS, 2018.

[3] Dongxian Wu and Yisen Wang. Adversarial Neuron Pruning Purifies Backdoored Deep Models. In NeurIPS, 2021.


**Summary Of The Paper:**

This paper proposes a measure to distinguish benign neurons and malicious neurons. Based on the measure, the authors identify the malicious neurons and update their parameters. Experimental results show the effectiveness of the proposed method.

**Summary Of The Review:**

This paper misses many details and is hard to understand. Even if the proposed method might be promising, the author should clarify it well. In my opinion, it requires a major revision. Thus, I recommend rejecting this paper.

---

> ### Author Response · Authors · 2022-11-16
> **Response (1/1)**
>
> Question 1: This paper is written badly and requires major revision. I have a terrible time understanding this paper. For example, when citing related works, please use brackets “(” \& “)” in most cases to include them, i.e., use \\citep in Latex.
>
> Response:
> We have revised this paper to make it more readable and clearer.
> Please see the revised version.
>
>
> Question 2: It is difficult to understand the formulation of Eqn (2).
>
> 1) Since DNN has an extremely large number of parameters, close-to-zero initialization still result in different loss compared to zero-initialization. Why could we $L_{D_c}(w_0)$ replace with $L_{D_c}(0)$?
>
> Response:
>
> We admit that close-to-zero initialization may result in different loss compared to zero-initialization.
> But, models with random close-to-zero initialization commonly predict randomly for inputs while models with zero-initialization also make similar predictions, thereby $L_{D_c}(w_0) \approx L_{D_c}(0)$.
>
> Moreover, the principle of defining BS is to evaluate the importance of a neuron to the contribution to the loss decreasing (model performance).
> In other words, we actually want to estimate the change in the performance of the model with and without a certain neuron.
> If the weights associated with the neuron are zero, the neurons can be considered to be removed.
> Therefore, $w_0$ is set to 0.
>
>
> 2) In this formulation, BS is defined for the whole model and every neuron share this value inside the model. So how can we distinguish neurons using this shared BS? Is the neuron-wise BS defined as the sum of the products of the parameters and their derivatives inside a specific neuron?
>
> We purify the input neurons in the last fully-connected layer.
> The BS of a specific neuron is defined as the sum of the negative product between the values and gradients of in-the-layer weights associated with the neuron.
> We have revised Section 4.1.1 to make our approach clearer.
>
> Question 3: Where is Figure 4 in Sec 5.4?
>
> Figure 4 is in Appendix A.2.
>
>
> Question 4: If this paper regards it as a pruning method, it neglects an important work, ANP [1]. Since this is another pruning-based method, this paper should compare it.
>
> We add ANP [1] as a baseline in the revised version.
> The following table reports the defense performance of ANP and WIPER and the overall performance of our method still outperforms ANP.
>
> |   Defense   | Before |       |    ANP    |          |   WIPER   |          |
> |:-----------:|:------:|:-----:|:---------:|:--------:|:---------:|:--------:|
> |    Attack   |   ACC  |  ASR  |    ACC    |    ASR   |    ACC    |    ASR   |
> |    BadNet   |  87.42 |  99.5 |   83.02   |   5.45   | **83.20** | **5.03** |
> |      BA     |  87.67 | 99.77 |   82.26   |   5.46   | **82.82** | **5.22** |
> |     ETA     |  87.2  | 97.24 |   81.20   |   6.45   | **82.41** | **5.80** |
> |      IA     |  86.83 | 98.67 |   80.91   | **5.78** | **82.04** |   6.02   |
> |     SIG     |  85.14 |  91.1 |   82.30   |   5.28   | **82.41** | **3.73** |
> |   TrojanNN  |  86.83 | 98.67 | **84.80** |   6.79   |   84.67   | **4.68** |
> |     IMC     |  87.10 | 99.78 |   80.04   |   7.43   | **80.92** | **6.77** |
> |   WaveNet   |  85.87 | 97.68 |   82.48   |   7.21   | **82.59** | **5.81** |
> |     SSA     |  86.30 | 98.21 |   80.48   |   6.54   | **81.70** | **4.76** |
> | LogitAttack |  86.92 | 99.98 | **83.09** |   7.65   |   83.03   | **5.54** |
>
> [1] Wu, Dongxian and Yisen Wang. “Adversarial Neuron Pruning Purifies Backdoored Deep Models.” ArXiv abs/2110.14430 (2021): n. pag.

---

### Official Review · Reviewer_pBHg · 2022-10-25

**Confidence:** 4
**Clarity, Quality, Novelty And Reproducibility:** Clarity, Quality, Novelty And Reprodu…
**Correctness:** 3
**Technical Novelty And Significance:** 3
**Empirical Novelty And Significance:** Not applicable
**Recommendation:** 5

**Strength And Weaknesses:**

Strong Points:

1. A novel a gradient-based detection method of bad neurons.
2. The method focused on the last fully connected layers to remove backdoor.
3. A new Adaptive Regularization (AR) mechanism is proposed to assist in purifying these identified bad neurons via fine-tuning.
4. Extensive experiment results illustrated the good performances.

Weak point:

There is a paper published at CCS 2021: “AI-Lancet: Locating Error-inducing Neurons to Optimize Neural Networks”. This paper also considered gradient during finding bad neurons. They also achieved good results with respect to accuracy and ASR. Authors have not mentioned this paper nor compared with. They should include this paper for comparison to prove the effectiveness of their method.


**Summary Of The Paper:**

The authors proposed a novel backdoor defense, which used first-order gradient to identify bad neurons using clean dataset. Also, a new Adaptive Regularization (AR) mechanism is used to fine-tune the model using detected bad neurons. They performed experiments on ten different backdoor attacks with three benchmark datasets. Their proposed method has performed better than other six state-of-the-art defense methods.

**Summary Of The Review:**

The authors used clean data to detect bad neurons and designed a new Adaptive Regularization (AR) mechanism for finetuning. However, they failed to mention and compare a very similar approach published in CCS’21, which also achieved good clean accuracy, ASR. Since these two algorithms are belongs to the same category. Lacking this comparison makes its contribution not clear.

---

> ### Author Response · Authors · 2022-11-16
> **Response (1/1)**
>
> Question:
> There is a paper published at CCS 2021: “AI-Lancet: Locating Error-inducing Neurons to Optimize Neural Networks”. This paper also considered gradient during finding bad neurons. They also achieved good results with respect to accuracy and ASR. Authors have not mentioned this paper nor compared with. They should include this paper for comparison to prove the effectiveness of their method.
>
>
> Response:
> Thanks for your comment and we add [1] in the revised paper as a baseline (Appendix A.2).
> The results in CIFAR-10, the following table, show that for some state-of-the-art attacks (BA, ETA, and SSA), the defense effect of our method outperforms AI-lancet a lot.
> The reason is as follows.
> | Defense | AI-Lancet |       |   WIPER   |          |
> |:-------:|:---------:|:-----:|:---------:|:--------:|
> |  Attack |    ACC    |  ASR  |    ACC    |    ASR   |
> |  BadNet |   80.15   |  5.21 | **83.20** | **5.03** |
> |    BA   |   73.81   | 27.17 | **82.82** | **5.22** |
> |   ETA   |   68.67   | 29.93 | **82.41** | **5.80** |
> |   SSA   |   68.47   | 22.14 |  **81.70** | **4.76** |
>
> Similar to our defense, AI-lancet consists of two stages: 1) locating bad neurons; 2) cleansing these neurons via fine-tuning.
> The difference is that AI-lancet finds bad neurons by measuring the behaviour change of the target model as being input with triggered data and clean data whose trigger regions are removed.
> Such an idea works well for pixel-level, or picture-level, backdoor attacks, however, fails to  defend state-of-the-art attacks whose triggers are usually crafted noises and can cover the whole image.
> In contrast, our method dose not suffer from this drawback because our neuron purifying strategy starts from the perspective of neurons themselves but not specific inputs.
>
> [1] Zhao, Yue et al. “AI-Lancet: Locating Error-inducing Neurons to Optimize Neural Networks.” Proceedings of the 2021 ACM SIGSAC Conference on Computer and Communications Security (2021): n. pag.

---

### Author Response · Authors · 2022-11-16
**Updates in the revision**

We thank all reviewers for their constructive comments. We have revised the paper to address the reviewers' concerns. The revised parts are in red.
The major changes include:

1. Two extra baselines were added. We showed the vulnerability of AI-Lancet against state-of-the-art attacks (Appendix A.1). We compared our method with ANP (Table 1). We also compared BS with other evaluation metrics for neuron importance, such as integrated gradient (Appendix A.1).

2. We revised the description of Section 4 (Approach) to make it clearer. We clarify how to define BS of a certain neuron and why $L_{D_c}(w_0) \approx L_{D_c}(0)$.

3. We improved the writing of the paper. We added a global figure (Figure 1) to overview our approach.  We elaborated on the discussion about L1 and L2 regularizations. We also fixed some typos citation issues, etc.

---

### Decision · Program_Chairs · 2023-01-20

**Decision:**

Reject

**Justification For Why Not Higher Score:**

The lack of several related and important related work downgrade the novelty and effectiveness of the proposed method, as well as the completeness of the manuscript. The writing is very poor. No study of the adaptive attack.

**Justification For Why Not Lower Score:**

N/A

**Metareview: Summary, Strengths And Weaknesses:**

This work proposed a new backdoor defense method by identifying bad neurons in a backdoor model, according to a new metric called benign salience.

The main concerns from the reviewers include:
1. Lack of important related methods: 1) the work with similar idea published in CCS’21, which also utilized gradient to identify bad neurons; 2) various methods that measure the contribution of neuron; 3) some advanced backdoor defense methods, like ANP (and several other advanced methods). Although the authors added partial evaluations, they are insufficient to verify the superior performance of the proposed method. The lack of so many related works from different perspectives reveals that the work was not well prepared before submission.
2. The writing is very poor.

Moreover, the behind assumption of identifying bad neurons is that the neurons could be separated into good and bad neurons. However, the adaptive attack could produce a backdoor model within which the activation paths of both poisoned and clean samples are largely overlapped, such that the proposed method is expected to be evaded. It should also be discussed and studied.